# Performance Evaluation of a Novel Combination of Four- and Five-Carbon [Butyric and Valeric] Short-Chain Fatty Acid Glyceride Esters in Broilers

**DOI:** 10.3390/ani14040617

**Published:** 2024-02-14

**Authors:** Marta I. Gracia, Patricia Vazquez, Yolanda Ibáñez-Pernía, Jeroen Pos, Snehal Tawde

**Affiliations:** 1Imasde Agroalimentaria, S.L. C/Nápoles 3, 28224 Pozuelo de Alarcón, Spain; pvazquez@e-imasde.com (P.V.); yipernia@e-imasde.com (Y.I.-P.); 2Perstorp Animal Nutrition, Perstorp Waspik BV, 5165 NH Waspik, The Netherlands; jeroen.pos@perstorp.com (J.P.); snehal.tawde@perstorp.com (S.T.)

**Keywords:** gut health, short-chain fatty acids, Butyric, Valeric, performance, feed conversion ratio, body weight, European poultry efficiency factor

## Abstract

**Simple Summary:**

A novel combination of Butyric and Valeric acid glycerol esters with oregano oil in a dry powder form was evaluated for performance improvements in broilers. The dosing regimen (500 g/Ton feed in starter and grower; 250 g/Ton in finisher feed) was considered low compared to conventional practices using non-esterified Butyric and Valeric short-chain fatty acids (SCFA). Six trials were conducted at various trial facilities in Italy, United Kingdom, Spain, and Poland. Supplemented broilers grew more (66.4 vs. 64.5 g/day; *p* = 0.0005), ate more feed (104.7 vs. 103.1 g/day; *p* = 0.0473), exhibited significantly lower feed conversion (1.58 vs. 1.60; *p* = 0.0072), leading to better EPEF value (410 vs. 389; *p* = 0.0006) than the control broilers. Meta-analysed trial performance data for novel SCFA formulations such as these are not commonly available, and serve to facilitate efficacy determination from an end-user perspective. The use of short- and medium-chain fatty acid esters in optimal low-dose combinations to reliably augment gut health and performance appears promising in commercial broiler production, and may lead to further improvements in industry practices and reduced antibiotic use.

**Abstract:**

A novel combination of Butyric and Valeric acid glycerol esters with oregano oil in a dry powder form was evaluated for performance improvements in broilers. The dosing regimen (500 g/Ton feed in starter and grower; 250 g/Ton in finisher feed) was considered low compared to conventional practices using non-esterified Butyric and Valeric short-chain fatty acids (SCFA). Six trials were conducted at various trial facilities in Italy, United Kingdom, Spain, and Poland. Supplemented broilers weighed significantly more than the control birds at 28 days of age (+3.4%; 1459 g vs. 1412 g; *p* = 0.0006) and at 42 days of age (+2.5%; 2834 g vs. 2763 g; *p* = 0.0030). Supplementation significantly reduced mortality from 1.9% to 0.8% during the finisher phase (from 29 to 42 days of age); however, average mortality was 3.2% for the whole 42-day growth period and was not affected. Further, supplemented broilers grew more (66.4 vs. 64.5 g/day; *p* = 0.0005), ate more feed (104.7 vs. 103.1 g/day; *p* = 0.0473), converted feed significantly more efficiently (1.58 vs. 1.60; *p* = 0.0072), leading to better EPEF value (410 vs. 389; *p* = 0.0006) than the control broilers. Meta-analysed trial performance data for novel SCFA formulations such as these are not commonly available, and serve to facilitate efficacy determination from an end-user perspective. The use of short- and medium-chain fatty acid esters in optimal low-dose combinations to reliably augment gut health and performance appears promising in commercial broiler production, and may lead to further improvements in industry practices and reduced antibiotic use.

## 1. Introduction

Gut-health linked feed additives occupy an established space in broiler feed formulations, their application being justified from a preventive and holistic approach. Rising costs associated with raising chicken, including feed costs necessitate that said feed additives not only deliver on their mechanistic promise, but also can be evaluated in terms of performance gains linked to better health and immunity. Even if the biochemical, physiological, and mechanistic pathways of several feed additives are well-understood, performance evaluation data under controlled conditions, using robust statistical processing, are lacking.

Owing to the increasingly shortening growth periods of the modern broiler, some of their inherent biological and physiological processes cannot function at a level that would otherwise be possible in nature, like a backyard/heritage chicken. These can include the establishment of a healthy gut microflora, and optimal digestive processes, etc. The need to rapidly synthesize lean muscle puts additional demands on these young birds, while being housed in high-density housing often in the presence of multiple pathogenic microorganisms. Antibiotics have been used for a while as a quick fix under these challenging conditions, leading to issues like antibiotic resistance. That is increasingly being phased out due to legislation, or consumer demand. In such a complex scenario, supplementation with additives that help to build immunity, better health and efficient growth may be helpful. The decision-making process will be more pragmatic if stakeholders can obtain access to bird performance datasets with robust statistical analyses.

Feed supplementation can broadly be classified under two categories: non-native/unknown-to-the-bird substances, or native molecules produced by the birds themselves. Butyric acid (BA), a four-carbon short-chain fatty acid (SCFA), is among the supplemented native gut-health enhancing molecules. It is naturally produced by microbial fermentation in gastrointestinal systems and is considered a primary nutrient providing energy to colonocytes. It is also a signalling molecule involved with gut-centric gene expression, cell differentiation, tissue development, immune modulator, and oxidative stress reducer [1]. BA is a preferential energy source which is oxidized in priority compared to other energy sources [2].

During the last two decades, BA, and its esters, salts and coated forms, have been commonly employed globally as a feed supplement in poultry production. The addition of BA salts and esters to turkey diets significantly improved the feed conversion ratio (FCR), increased the values of the European Poultry Efficiency Factor (EPEF) and duodenal villus height, and decreased the faecal populations of *Escherichia coli* and *Clostridium perfringens* [3]. BA has been observed to play a role in maintaining the performance and carcass quality of vaccinated broilers challenged in coccidiosis [4]. The administration of BA in vitro at 0.8, 1, and, 1.2% induced a dose-dependent inhibition of sporulation and damage of coccidian oocysts, while in vivo evaluation was correlated with improved weight gain and FCR, with concomitant reductions in lesion and oocyst score and oocyst per gram of faeces [5]. In a dietary energy and amino acid reduction study, BA improved broiler performance and modulated the cecal microbiota and the immune response [6].

Valeric acid, a five-carbon SCFA, is also produced naturally by specific members of the microbiota of the lower intestinal tract of chicken and other species. VA is, however, present at a lower concentration than BA in the intestinal tract. Unlike BA, very little is known about the effects of Valeric acid on intestinal health and on health in general. Esters of BA and VA have been suggested to have a preventive role in lipid disorders, impaired intestinal barrier function and inflammation caused by high-fat diets and genetic predisposition in rats [7]. VA is associated with rapid gut microbiota maturation in humans [8]. BA and VA were individually tested for direct antimicrobial activity, and at sub inhibitory concentrations were found to significantly reduce the motility and biofilm formation in *Salmonella* strains isolated from poultry [9]. Irradiated mice treated with VA showed improved survival rates, better protected hematogenic organs, and improved gastrointestinal tract function and intestinal epithelial integrity [10]. Both BA and VA glycerol esters, were separately investigated for their effect on intestinal health, in a necrotic enteritis challenge model, with results comparable to bacitracin [11]. However, the high doses needed for the effect resulted in reduced feed intake and growth, which is a side-effect that needs further investigation. Further, VA monoester applied at 1.5 kg/t provided significant control of both clinical and subclinical NE and produced the lowest levels of mortality by NE and total mortality of the challenged groups [11]. Onrust et al. [12] also observed that VA supplementation to broiler feed can reduce the incidence of necrotic enteritis, also finding a decrease in FCR and positive effect in the morphology of the small intestinal mucosa.

Oregano oil (OO) is made using a distillation-based production process that concentrates the active ingredients like carvacrol and thymol, among other components. Peng et al. [13] found that the use of OO had a positive influence over the intestinal health, and thereby improved the growth performance and carcass traits of broilers. Previously, a combination of BA and the 12-carbon containing lauric acid with microencapsulated essential oils containing thymol, cinnamaldehyde and eucalyptus had been found to reduce the severity of Necrotic Enteritis lesions [14]. The anti-inflammatory properties of OO [15,16] together with its antimicrobial, immunomodulatory and performance enhancing characteristics [17] were considered a promising adjunct to the properties of BA and VA described earlier. It was hypothesized that using low-dose BA, VA, and OO supplementation would enhance broiler gut health and performance. The goal of this study was to evaluate the said BA, VA, and OO combination in different country/production scenarios with a focus on animal performance parameters.

## 2. Materials and Methods

Six experiments were carried out between 2019 and 2021 at trial sites in Italy (experiment one), UK (experiments two and five), Spain (experiments three and six), and Poland (experiment four) to assess the potential efficacy, and performance of a novel feed additive containing Glycerol esters of BA and VA, with Oregano oil (hereafter described as BVA, for brevity) in broiler diets (Table 1). The rationale behind the test formulation was to improve the performance of birds further by augmenting the qualities of BA, with the next-in-line five-carbon containing SCFA. VA was used in lower proportion to BA to mimic their ratio’s observed chicken gastrointestinal systems. Previously, unpublished trials of individual BA and VA glyceride esters served to determine the low-dose regimen used in this study. Inclusion of OO was based on its activity spectrum and the ability to include it at lower dose while maintaining a threshold activity potential. In each of the six experiments, male broilers (Ross 308) were placed in floor pens and were fed a starter feed from 0 to 14 days of age, a grower feed from 15 to 28 days of age, and a finisher feed from 29 to 42 days of age. All experiments used mash feeds except experiment 4 employing pelleted diets during the grower and finisher phases. The experimental diets were fed ad libitum and were based on wheat/corn/soybean meal (rapeseed meal also in experiments 3, 4, and 6) with no added coccidiostat (except experiment 4 that included Coxidin (Coxidin^®^ 200 Microgranulate, Huvepharma)), growth promoter or veterinary antibiotic (Table 2).

In each experiment, the birds were divided into two groups, where each received one of two experimental treatments; T1 was the control basal diet (Table 2) and T2 was the same diet formulation supplemented with 500 mg/kg in the starter and grower feeds and 250 mg/kg in the finisher feeds of BVA. For each feeding period (starter, grower, and finisher) in all six experiments, both experimental diets consisted of the same basal diet, and met or exceeded the nutrient requirements recommended by NRC [18] for broilers.

In experiment one, 500 day-old chicks were used (250 for Control and 250 for BVA) and housed in pens of 25 broilers (10 pens per treatment). In experiment two, 528 day-old chicks were used (308 for Control and 220 for BVA) and housed in pens of 22 broilers (14 pens per treatment for control, 10 for BVA). In experiment three, 616 day-old chicks were used (352 for Control and 264 for BVA) and housed in pens of 22 broilers (16 pens per treatment for control, 12 for BVA). In experiment four, 1200 day-old chicks were used (600 per treatment) and housed in pens of 60 broilers (10 pens per treatment). In experiment five, 280 day-old chicks were used (140 for Control and 140 for BVA) and housed in pens of 14 broilers (10 pens per treatment). In experiment six, 396 day-old chicks were used (198 per treatment) and housed in pens of 22 broilers (9 pens per treatment).

Each trial was conducted to meet legislation applying to commercial production at the time and European Ethical and Welfare standards for broiler chickens. The husbandry of birds during the experiments was comparable to that which is practised commercially in the EU, with feed and water supplied ad libitum. Total pen bird weights and feed intake were measured on days 0, 14, 28, and 42. Mortality was measured daily and collected. The buildings were supplied with artificial, programmable lights, automated electric, biomass, or gas heating and forced ventilation. The temperature was maintained at 32 to 35 °C at the start of the experiment and decreased by 3 °C each week. From day 28 until the end of the experiment (day 42), the temperature was set to 21–23 °C. The lighting programme was 23 h light and 1 h dark during each 24 h period throughout the experiment (experiments 2 and 5), and 23 h light and 1 h dark for first 7 days, 18 h light and 6 h during each 24 h period until the end of study (experiments 1, 3, 4, and 6).

Data were tested for homogeneity between experiments and then pooled to enable statistical analysis of the whole dataset from the six experiments using the GLMM procedure of SPSS version 28.0.1.1 [19] with BVA supplementation as fixed effect and experiment as random effect. Probabilities of *p* ≤ 0.05 were significant and 0.05 < *p* ≤ 0.10 were a near-significant trend. The data analysed included body weight (BW), mortality, average daily gain (ADG), average daily feed intake (ADFI), feed conversion ratio (FCR), and European Poultry Efficiency Factor (EPEF). EPEF was calculated using the following equation:EPEF = (ADG (g)/FCRx10) × (100 − % mortality).

## 3. Results

The difference between broilers fed the BVA treatment and those fed control diet was investigated for the main global (0–42 d) parameters (BW, ADFI, FCR, and EPEF). Forest plot description of study comparisons showed that among all trials, broilers supplemented with BVA showed higher BW, feed intake and EPEF in 4 of 6 trials and lower FCR in 3 of 6 trials, although significance was only reached in one trial for BW, ADFI, and FCR and in three trials for EPEF (Figure 1). No adverse events or incidence that affected the animals clinically were recorded in any of the trials.

The effect of BVA treatment on BW and mortality is shown in Table 3. Broilers fed BVA weighed significantly more than the control chicks at 28 days of age (+3.4%; 1459 g vs. 1412 g; *p* = 0.0006) and at 42 days of age (+2.5%; 2834 g vs. 2763 g; *p* = 0.0030). The average BW of broilers at 42 days of age ranged from 2.44 kg in experiment two to 3.07 kg in experiment six, all of which were within the expected range of commercial weights at this age. BVA supplementation significantly reduced mortality from 1.9% to 0.8% during the finisher phase (from 29 to 42 days of age). Average mortality was 3.2% for the whole 42-day growth period and was not affected by treatment.

The influence of BVA supplementation on ADG, ADFI, FCR, and EPEF by growth periods is summarised in Table 4. For the global performance period (from 0 to 42 days of age) broilers fed the BVA supplemented diets grew more (66.4 vs. 64.5 g/day; *p* = 0.0005), ate more feed (104.7 vs. 103.1 g/day; *p* = 0.0473), exhibited significantly lower FCR (1.58 vs. 1.60; *p* = 0.0072), resulting in better EPEF value (410 vs. 389; *p* = 0.0006) than the control broilers.

During the starter phase, BVA supplementation significantly improved ADG (27.9 vs. 27.3 g/day; *p* = 0.0435). During the grower phase, BVA supplementation significantly increased ADG (71.8 vs. 69.1 g/day; *p* = 0.0005) and feed intake (105.7 vs. 103.8 g/day; *p* = 0.0473), and reduced the FCR (1.48 vs. 1.51; *p* = 0.0091). During the finisher phase, BVA supplementation improved ADG and feed intake, a near-significant trend.

## 4. Discussion

Combining trials conducted in different countries (Italy, UK, Spain, and Poland) with corresponding differences in production practices (i.e., different diets/raw material combinations) allowed us to test the performance enhancing potential of BVA. It is not uncommon to find that experimental formulations that perform well in one country/production scenario fail to perform elsewhere. Further, overcoming this barrier involves wide testing, followed by a robust meta-analyses of the production data. Meta-analysed experimental trial data evaluating the efficacy of animal feed additives are not readily available in the public domain. A meta-analytical approach “offers the opportunity to critically evaluate and statistically combine results of comparable studies or trials. Its major purposes are to increase the numbers of observations and the statistical power, and to improve the estimates of the effect size of an intervention or an association” [20]. In addition to the Triple R [Reduce, Refine, Replace] initiative towards animal experimentation, increasing cost and risk-to-return ratio and rapid pace of global commodification limits the number of trials that can be conducted in a defined period. Lack of comparative, and standardized performance data therefore negates the statistical processing that is necessary to differentiate, whether a feed additive indeed delivers meaningful results to the producer, in terms of improved animal performance, or not. Indeed, while it may seem counter-intuitive in a predicted era of lab-grown meat and slow-growing chicken, a holistic performance-enhancing feed additive that augments gut health may likely find a place as a viable option in the toolbox of an intensive broiler producer.

A multitude of physiological, biochemical, immunological, and microbiomic data are now available for BA-based feed additives [2]; however, performance evaluation metadata sets supported by robust statistical processing are lacking. The authors note a significant drawback in numerous studies on BA-based feed additives in the frequently used high doses [≥1000 g/Ton], and in the lack of dose–response studies. From the combined statistical analyses of data from the six experiments, it was concluded that diet supplementation with BVA at 500 g/ton between 0 and 28 days of age and 250 g/ton between 29 and 42 days of age was effective in improving broiler performance. Broilers supplemented with BVA were significantly heavier at 42 days (2834 vs. 2763 g) compared to the control birds. Over the entire experimental period (0 to 42 days of age), broilers fed the supplemented diet grew 3% more, ate 1.5% more feed, were 1.25% significantly more efficient, and showed 5.7% higher EPEF than the control birds. Onrust et al. [12] similarly reported significant improvement in BW (3%) and in FCR (3.3%) with VA supplementation. Contrary, other authors reported lower feed intake [11,21] as compared to control, suggesting that although higher levels of the acids are more effective in controlling mortality and lesions, these levels may be associated with detrimental effects on feed intake and consequently weight gain. Mortality in the finisher phase (day 29–42) was significantly reduced, which could help to minimize losses as the producer spends more on feeding birds in this phase, compared to earlier phases. Such savings on feeding adds to the overall efficiency of the operation; which is reflected in the EPEF improvements. Except mortality, significant improvements in all parameters throughout the 42-day period, combined with ADG improvements in starter, and overall improvements in grower phase, indicate that the formulation is overall efficacious, especially during the early–mid growing phase. It could be hypothesized that the additional energy supplementation to the chicks colonocytes through BA esters, the antimicrobial effect of VA and the anti-inflammatory and associated effects of oregano oil combined, lead to chicks growing faster, eating more and thereby converting feed into gain more efficiently. As a result, birds in commercial production may reach market weight earlier and/or less production days, possibly higher protein yield, or less feed consumption for equivalent yield in less time. The combined multi-modal effect is also likely to be correlated with reduced finisher stage mortality. The risk and benefits associated with a higher dose than used in these trials are not well-characterized and need further investigation. A somewhat similar approach (combination of BA sodium salt with cinnamaldehyde and thymol, but lacking VA) was noted to significantly reduce Salmonella contamination in the cecum but could not enhance bird performance [22] even if higher doses were used. Performance improvements observed in this study could be associated with the combined, possibly synergistic and/or complementary mode of action of BA, VA, and OO. It was considered that BA provides the energy to colonocytes during the vital early growth stages and enabling improved gut-centric gene expression and tissue development [1], while VA provides the antimicrobial and anti-inflammatory impetus [9,10], supported by OO driving improved digestion and mucus production associated with relief from pathogen adhesion inhibition [23]. The combined histone deacetylase inhibition potential of BA and VA may also have contributed to their protective mechanism leading to better growth, wherein histone deacetylase overexpression is associated with a variety of disease pathologies ranging from colitis and cardiovascular disease to cancer [24]. It can be hypothesized that supplementation with the BA and VA ester SCFAs, in lower doses than published research, complemented with phytobiotic molecules (OO), can be an efficient and reliable method of protecting and enhancing broiler gut health, resulting in the improved performance observed in the current study. Performance datasets such as these could aid in making apple-to-apple comparisons with other approaches. With the limited data on the use of VA and VA esters available in the literature, further comparisons cannot be undertaken at the moment. Low-dose supplementation with a combination of BA and VA esters, with phytobiotics, is a less explored approach, and the present study contributes to the body of knowledge.

The combined analysis of the dataset indicated significant improvements in broiler performance over multiple trials in four countries, increasing the likelihood that such formulations could add value to poultry production over different geographical and production situations. Future possibilities might include the evaluation of varying combinations of BA and VA esters complemented with phytobiotic molecules under pathogen challenge models and on the mechanism of action in healthy animals, wherein the potential usefulness of VA esters used alone [12] serves as a precedent.

## 5. Conclusions

According to the results obtained from the combined analysis of different trials, it can be hypothesized that supplementation with the BA and VA ester SCFAs, in lower doses, complemented with phytobiotic molecules can be an efficient and reliable method of protecting and enhancing broiler gut health. The use of short- and medium-chain fatty acid esters in optimal low-dose combinations to reliably augment gut health and performance appears promising in commercial broiler production, and may lead to further improvements in industry practices and reduced antibiotic use.

## Figures and Tables

**Figure 1 animals-14-00617-f001:**
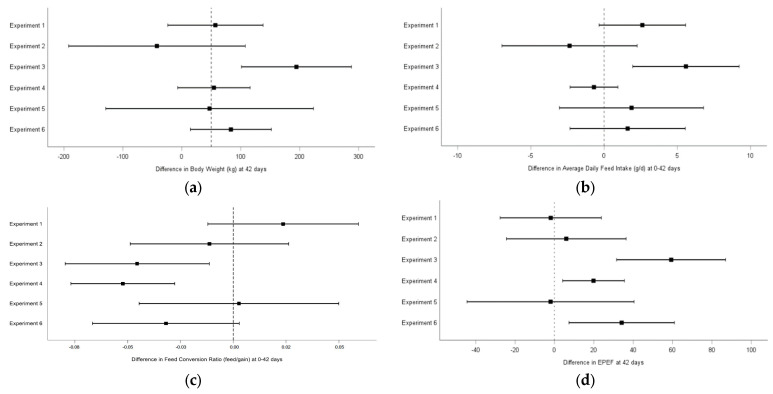
Effect sizes with 95% confidence intervals between BVA and Control treatments for body weight at 42 days (**a**), average daily feed intake at 0–42 days (**b**), feed conversion ratio at 0–42 days (**c**), and EPEF at 42 days (**d**).

**Table 1 animals-14-00617-t001:** Characteristics of the experiments included in the analysis.

Trial Information						
Experiment	Location	Replications per Treatment	Birds per Replication	Total Birds	Diets	Diet Type
1	Italy	10	25	500	Wheat/corn/soybean meal/full fat soybean/soybean oil	Mash
2	UK	14/10	22	528	Wheat/corn/soybean meal/full fat soybean/soybean oil	Mash
3	Spain	16/12	22	616	Corn/wheat/soybean meal/rapeseed meal/soybean oil/lard	Mash
4	Poland	10	60	1200	Wheat/corn/triticale/soybean meal/rapeseed meal/soybean oil	Mash/pellet
5	UK	10	14	280	Wheat/corn/soybean meal/full fat soybean/soybean oil	Mash
6	Spain	9	22	396	Corn/wheat/soybean meal/rapeseed meal/soybean oil/lard	Mash

**Table 2 animals-14-00617-t002:** Characteristics of the basal experimental diets used in the six experiments.

Experiment	Experiments 2 and 5			Experiments 3 and 6			Experiment 4			Experiment 1		
Days	0–14	14–28	29–42	0–14	14–28	29–42	0–14	14–28	29–42	0–14	14–28	29–42
Main ingredients												
Corn	-	25.0	-	30.0	32.0	32.0	10.0	10.0	10.0	42	44.3	40
Wheat	55.1	29.2	60.9	21.5	21.0	20.6	53.0	42.8	33.8	15	17.3	26.5
Triticale	-	-	-	-	-	-	-	10.0	20.0	-	-	-
Soybean meal	30.0	28.0	17.0	31.5	25.8	20.4	24.8	20.0	14.1	24	17.6	9.55
FFSB	8.0	10.0	15.0	-	-	-	-	-	-	4.46	9.56	12.51
Rapeseed meal	-	-	-	8.0	10.0	15.0	5.0	8.0	12.0	-	-	-
Soybean oil	2.4	4.0	3.4	4.8	1.5	1.5	3.5	5.8	7.3	1.35	1.51	2.59
Animal fat	-	-	-	-	6.0	7.4	-	-	-	-	-	-
Others	4.5	3.8	3.7	4.2	3.7	3.1	3.7	3.4	2.8			
Calculated nutritional content												
ME (kcal/kg)	3000	3099	3197	2950	3120	3190	2950	3075	3150	2975	3071	3213
Crude protein (%)	23.0	21.8	20.0	22.0	20.0	19.0	21.5	20.0	18.5	22.41	19.83	17.85
Digestible lysine (%)	1.28	1.16	1.06	1.27	1.10	0.97	1.35	1.16	1.00	1.41	1.25	1.06
Calcium (%)	0.96	0.88	0.81	0.96	0.88	0.78	0.85	0.83	0.75	0.77	0.65	0.57
Av. Phosphorus (%)	0.48	0.43	0.41	0.48	0.44	0.39	0.33	0.33	0.28	0.60	0.48	0.39

**Table 3 animals-14-00617-t003:** Effect of BVA supplementation on body weight (BW) and mortality for combined data from six broiler trials.

Parameter	Days of Trial	Treatment		SEM	*p*-Value
		T1 Control	T2 BVA		
BW, g	14	428	437	3.2	0.0623
	28	1412 ^b^	1459 ^a^	9.5	0.0006
	42	2763 ^b^	2834 ^a^	16.4	0.0030
Mortality, %	0–14	0.8	0.6	0.23	0.6012
	15–28	1.0	1.5	0.29	0.2978
	29–42	1.9 ^b^	0.8 ^a^	0.30	0.0150
	0–42	3.7	2.8	0.44	0.1606

Different superscripts (a, b) within a row indicate significant differences (*p* ≤ 0.05).

**Table 4 animals-14-00617-t004:** Effect of BVA supplementation on average daily gain (ADG), average daily feed intake (ADFI), feed conversion rate (FCR), and EPEF for combined data from six broiler trials.

Days of Trial	Parameter	Treatment		SEM	*p*-Value
		T1 Control	T2 BVA		
0–14	ADG, g/d	27.3 ^b^	27.9 ^a^	0.23	0.0435
	ADFI, g/d	35.8	36.3	0.26	0.2052
	FCR, feed/gain	1.32	1.31	0.009	0.1960
15–28	ADG, g/d	69.1 ^b^	71.8 ^a^	0.54	0.0005
	ADFI, g/d	103.8 ^b^	105.7 ^a^	0.64	0.0473
	FCR, feed/gain	1.51 ^b^	1.48 ^a^	0.009	0.0091
29–42	ADG, g/d	97.0	99.2	0.81	0.0529
	ADFI, g/d	170.6	173.3	1.13	0.0944
	FCR, feed/gain	1.77	1.76	0.008	0.2922
0–42	ADG, g/d	64.5 ^b^	66.4 ^a^	0.41	0.0011
	ADFI, g/d	103.1 ^b^	104.7 ^a^	0.55	0.0432
	FCR, feed/gain	1.60 ^b^	1.58 ^a^	0.005	0.0072
	EPEF	389 ^b^	410 ^a^	4.2	0.0006

Different superscripts (a, b) within a row indicate significant differences (*p* ≤ 0.05).

## Data Availability

The data presented in this study are available on request from the corresponding author. The data are not publicly available due to privacy concerns.

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
