# Peer review of "Performance Evaluation of a Novel Combination of Four- and Five-Carbon [Butyric and Valeric] Short-Chain Fatty Acid Glyceride Esters in Broilers"

_animals, 2024, doi:10.3390/ani14040617_

Round 1

Reviewer 1 Report

Comments and Suggestions for Authors

Comments to the Authors of manuscript number: animals-2841301 entitled “Performance evaluation of a novel combination of 4 and 5-Carbon [Butyric and Valeric] short chain fatty acid glyceride esters in broilers”.

A novel combination of Butyric and Valeric acid glycerol esters with oregano oil in a dry powder form was assessed for enhancing broiler performance, with a dosing regimen (500 g/Ton feed in starter & grower; 250 g/Ton in finisher feed) considered low compared to conventional practices. Six trials conducted across different facilities demonstrated that supplemented broilers exhibited increased growth, higher feed consumption, more efficient feed conversion, and better EPEF values compared to control birds. The supplemented broilers also showed reduced mortality during the finisher phase. Meta-analyzed trial data for such novel SCFA formulations are scarce but provide valuable insights for end-users. The use of short and medium chain fatty acid esters in optimal low-dose combinations holds promise for improving gut health and performance in commercial broiler production, potentially contributing to enhanced industry practices and reduced antibiotic use.

1. the introduction is very long, but I like it.

2. There's noticeable repetition in emphasizing the role of BA and VA. Streamline the information, avoiding redundancy to maintain reader engagement.

3. Some sentences are overly complex and may impede understanding.

4. L 115-133- this part should be rephrased. Here should be presented a clear hypothesis without the description of the preparation of factors assessed in the study, and then the goal of the study should be presented.

5. L 135-138 why did this study involve so many countries? It is unclear?

6. L 152 – where are these diets presented? The table lacks these description

7. Were these factors given on top? How the isonutrition of the diet were kept?

8.  nowhere is it mentioned how much each of the additives constituted in each of these diets. Each diet should be presented well. Consider providing a brief justification for the chosen dosage of BVA in the experimental diets. Explaining how this dosage was determined or its relevance to existing literature would add depth to the methodology.

9. While the husbandry practices align with commercial standards, providing more specific details on the standardization of procedures, such as how feed and water were monitored, could enhance the reproducibility of the study.

10. L 207 – the measurement of mortality is not explained in material and methods

11. Explicitly state the rationale behind conducting trials in multiple countries (Italy, UK, Spain, and Poland). Discuss how these diverse environments contribute to the robustness of the study and highlight any potential variations due to different production practices.

12. Reiterate the justification for selecting the specific BVA dosage (500 g/ton between 0-28 days and 250 g/ton between 29-42 days). Provide reasoning for this dosing strategy in relation to previous studies or established standards.

13. Emphasize any findings related to mortality in the results. If there's a significant reduction in mortality during the finisher phase, its potential implications and benefits should be discussed more explicitly.

14. Further elaborate on the potential synergistic effects of BA, VA, and OO. Discuss how their combined action may contribute to the observed improvements in broiler performance. Providing a detailed mechanism of action can strengthen the scientific argument.

15. Acknowledge the need for further investigation into the potential risks and benefits associated with a higher BVA dose. Clearly outline the gaps in knowledge and the specific areas that warrant future research.

16. Discuss the findings in relation to previous studies, especially those using BA or VA. Highlight the uniqueness of the present approach

17. While presenting the significant improvements in broiler performance, provide context by comparing these results with industry standards or benchmarks.

18. The conclusion presents a well-structured and clear summary of the findings from the combined analysis of trials investigating the supplementation of broiler diets with Butyric Acid (BA) and Valeric Acid (VA) ester Short Chain Fatty Acids (SCFAs).

Author Response

Many thanks for the positive comments and all your help for improving the manuscript. We have taken into account all your suggestions.

  1. the introduction is very long, but I like it. Many thanks for the positive observation
  2. There's noticeable repetition in emphasizing the role of BA and VA. Streamline the information, avoiding redundancy to maintain reader engagement. Many thanks for the observation. We have reviewed, and edited the paragraphs for brevity/redundance.
  3. Some sentences are overly complex and may impede understanding. Many thanks for the observation. We have rephrased some sentences.
  4. L 115-133- this part should be rephrased. Here should be presented a clear hypothesis without the description of the preparation of factors assessed in the study, and then the goal of the study should be presented. Done
  5. L 135-138 why did this study involve so many countries? It is unclear? The experimental design involved a combination of several individual trials in different countries in order increase the robustness of the conclusions on the efficacy of the additive.
  6. L 152 – where are these diets presented? The table lacks these description. Many thanks for the observation. We have substituted the term diet by experimental treatment and added the term “basal” in order to indicate that the same basal diets were used without (T1) or with (T2) the additive.
  7. Were these factors given on top? How the isonutrition of the diet were kept? Many thanks for the observation. Yes, the BVA was added on top and the theoretical nutritional contribution was not taken into account. Isonutrition has been changed by the explanation that the same basal diet was used.
  8. nowhere is it mentioned how much each of the additives constituted in each of these diets. Each diet should be presented well. Consider providing a brief justification for the chosen dosage of BVA in the experimental diets. Explaining how this dosage was determined or its relevance to existing literature would add depth to the methodology. A justification on the design of the final formulation has been included into the M&M section, at L129-L135. The tested combination and dose was decided and based on previous internal unpublished trials.
  9. While the husbandry practices align with commercial standards, providing more specific details on the standardization of procedures, such as how feed and water were monitored, could enhance the reproducibility of the study. We do not clear understand what additional details to be added: type of feeders/drinkers? Water was administered via nipple drinkers ad lib and feed was ad lib available, and was weighed at the beginning and end of each feeding phase.
  10. L 207 – the measurement of mortality is not explained in material and methods Many thanks for the observation. The statement “Mortality was measured daily and collected” has been included in the M&M section. The fact that mortality was statistically analysed and used for calculating EPEF was already included in the corresponding section.
  11. Explicitly state the rationale behind conducting trials in multiple countries (Italy, UK, Spain, and Poland). Discuss how these diverse environments contribute to the robustness of the study and highlight any potential variations due to different production practices. The robustness is reached by the design of the trial as a meta-analysis. The difference in diet composition is now highlighted.
  12. Reiterate the justification for selecting the specific BVA dosage (500 g/ton between 0-28 days and 250 g/ton between 29-42 days). Provide reasoning for this dosing strategy in relation to previous studies or established standards. A justification on the design of the final formulation has been included into the M&M section, at L129-L135. The tested combination and dose was decided and based on previous internal unpublished trials. The reference to the selection of a lower dose than what is normally used in other publications is also included.
  13. Emphasize any findings related to mortality in the results. If there's a significant reduction in mortality during the finisher phase, its potential implications and benefits should be discussed more explicitly. Many thanks for the observation. We are sorry but we do not understand the comment. The findings related to mortality were described in the results (L.207-209). Also the significant and numerical reduction in mortality was discussed (L.262-277).
  14. Further elaborate on the potential synergistic effects of BA, VA, and OO. Discuss how their combined action may contribute to the observed improvements in broiler performance. Providing a detailed mechanism of action can strengthen the scientific argument. L280-L284 addresses the same, Thank you. Further elaboration was considered outside the scope of the publication, and was not undertaken to reduce speculation, in absence of sufficient scientific literature to support the said assumption.
  15. Acknowledge the need for further investigation into the potential risks and benefits associated with a higher BVA dose. Clearly outline the gaps in knowledge and the specific areas that warrant future research. In addition to the statement in L300-L302, the suggestion is now addressed in L104-L106
  16. Discuss the findings in relation to previous studies, especially those using BA or VA. Highlight the uniqueness of the present approach. The discussion has been increased including references to previous studies.
  17. While presenting the significant improvements in broiler performance, provide context by comparing these results with industry standards or benchmarks.
  18. The conclusion presents a well-structured and clear summary of the findings from the combined analysis of trials investigating the supplementation of broiler diets with Butyric Acid (BA) and Valeric Acid (VA) ester Short Chain Fatty Acids (SCFAs). Many thanks for the positive observation.

Reviewer 2 Report

Comments and Suggestions for Authors

Manuscript titled "Performance evaluation of a novel combination of 4 and 5-Carbon (Butyric and Valeric) short chain fatty acid glyceride esters in broilers."

Comments to the Author

The study is interesting, well planned and well written, and more importantly is based on innovative idea. The results of this study could impact the future trend in poultry farming. In my opinion, it is publishable. Some concerns to be addressed as follows:

Abstract:

Abstract is well written and results are well described.

Introduction:

Introduction is in line with the need of the study, but study objective is missing. Incorporate the study objective at the end of introduction.

Materials and methods:

Line 139-140: “In each of the six experiments, replicated floor pens of male broilers (Ross 308) were fed” Rewrite this sentence to make it more understandable.

Line 141-142: Why different forms (mash and pellet) of feed were used in different experiments? In my opinion, feed forms may affect feed consumption, and ultimately performance of the birds.

Line 143: Write it (ad-libitum) in italic “ad-libitum”.

Line 144: Why rapeseed meal was used in experiments 3, 4 and 6, and not in other experiments (1, 2, 5).

Line 144: Is Coxidin a coccidiostat? Why it was used only in experiment 4 and not in others?

Line 154-155: "For each feeding period (starter, grower, and finisher) in all six experiments, both diets were calculated to be isonutritive, and met or exceeded the nutrient requirements recommended by NRC [18] for broilers." It is not correct to use NRC 1994 recommendations for the nutrition of Ross 308 broilers in 2021, as their performance and nutrient requirements have changed significantly over the past 30 years.

Line 170: Write it (ad-libitum) in italic “ad-libitum”.

Line 173-174: "The temperature was maintained at 32 to 35°C at the start of the experiment and decreased by 3°C each week." What was the reason for not maintaining the temperature constant at the start of the experiment?

Line 175-178: I could not understand why different lighting programme was used in different experiments?

Line 193-194: Did you observe birds clinically? If yes, then what clinical parameters were observed?

In the discussion section, please elaborate putative mechanism of synergistic effects observed in the combined supplementation of short-chain fatty acids with organo oil.

Author Response

Many thanks for the recommendation for publication and for all your help for improving the manuscript. We have taken into account all your suggestions.

Abstract:

Abstract is well written and results are well described. Thanks for the observation.

Introduction:

Introduction is in line with the need of the study, but study objective is missing. Incorporate the study objective at the end of introduction. Done, L123-L124

Materials and methods:

Line 139-140: “In each of the six experiments, replicated floor pens of male broilers (Ross 308) were fed” Rewrite this sentence to make it more understandable. Done.

Line 141-142: Why different forms (mash and pellet) of feed were used in different experiments? In my opinion, feed forms may affect feed consumption, and ultimately performance of the birds. Thanks for the observation. Yes, feed form affect performance. Trials were designed to be as homogeneous as possible throughout the experimental sites. However, some differences were caused by local husbandry practices, that were not practically avoidable. That is the case of trial 4, in which mash feed was used at the beginning and pellets thereafter. However, we also think that the differences in management increased the robustness of the study.

Line 143: Write it (ad-libitum) in italic “ad-libitum”. Done.

Line 144: Why rapeseed meal was used in experiments 3, 4 and 6, and not in other experiments (1, 2, 5). Diet formulations were adapted to local availability and use of raw materials.

Line 144: Is Coxidin a coccidiostat? Why it was used only in experiment 4 and not in others? Yes, it is a coccidiostat. Again, diet formulations were adapted to local customs. As coccidiostats are normally used under commercial conditions, its use was considered as representative of standard production practices.

Line 154-155: "For each feeding period (starter, grower, and finisher) in all six experiments, both diets were calculated to be isonutritive, and met or exceeded the nutrient requirements recommended by NRC [18] for broilers." It is not correct to use NRC 1994 recommendations for the nutrition of Ross 308 broilers in 2021, as their performance and nutrient requirements have changed significantly over the past 30 years. Thanks for the observation. Yes, nutrient requirements of broilers have changes significantly. However, nutrient recommendations of genetic are not strictly followed commercially. That is why we used NCR as a “minimum” reference, however dig amino acids used in the diets are well above.

Line 170: Write it (ad-libitum) in italic “ad-libitum”. Done

Line 173-174: "The temperature was maintained at 32 to 35°C at the start of the experiment and decreased by 3°C each week." What was the reason for not maintaining the temperature constant at the start of the experiment? Although facilities were environmentally controlled, slight temperature fluctuations occur. Even, the start temperatures were not exactly the same in the whole six facilities. In order to simplify the M&M section, the range 32-35 was used, as it covers all the fluctuations, all the trials and the needs for the birds.

Line 175-178: I could not understand why different lighting programme was used in different experiments? Lighting programme was very similar between the 6 experimental trials. The differences corresponded to the local practices of each facility.

Line 193-194: Did you observe birds clinically? If yes, then what clinical parameters were observed? As no challenges were applied, no specific clinical parameters were evaluated. Daily evaluation of birds was done in order to identify any general health observation like diarrhoea, respiratory problems, leg problems, etc.

In the discussion section, please elaborate putative mechanism of synergistic effects observed in the combined supplementation of short-chain fatty acids with oregano oil. L280-L284 addresses the same, Thank you. Further elaboration was considered outside the scope of the publication, and was not undertaken to reduce speculation, in absence of sufficient scientific literature to support the said assumption.

Round 2

Reviewer 1 Report

Comments and Suggestions for Authors

no more comments